# STRUCTURAL PROBING WITH FEATURE INTERACTION

## ABSTRACT

Measuring nonlinear feature interaction is an established approach to understanding complex patterns of attribution in many models. In this paper, we use Shapley Taylor interaction indices (STII) to analyze the impact of underlying data structure on model representations in a variety of modalities, tasks, and architectures. Considering linguistic structure in masked and auto-regressive language models (MLMs and ALMs), we find that STII increases within idiomatic expressions and that Transformer ALMs scale STII with syntactic distance, just as LSTM-based ALMs do. Our speech model findings reflect the phonetic principal that the openness of the oral cavity determines how much a phoneme's acoustics vary based on context. Our wide range of results illustrates the benefits of interdisciplinary work and domain expertise in interpretability research.

## 1 INTRODUCTION

Feature attribution is a common approach to interpreting modern neural networks. One classic method is the Shapley decomposition (Shapley, 1952), which is adapted from game theory scenarios where the goal is to attribute credit—a **Shapley value**—within the context of a multi-agent cooperative game. The Shapley decomposition assumes approximately linear features, usually an incorrect assumption where deep learning is concerned. Consequently, researchers have often tried to quantify the validity of the approximate linearity assumption in Shapley values (Kumar et al., 2021). These methods calculate the degree of nonlinearity, or **Shapley interaction**, among a set of features; itself a metric that can be used to interpret the representations produced by a complex model.

This paper investigates Shapley interactions in a number of tasks and architectures. We use Shapley interactions as a case study to illustrate the importance of grounding model interpretations in the underlying structure of the data and the target models. To this end, we draw connections between interaction metrics and various structural properties of the data in each setting: syntax, tokenization, and idiomatic expressions in masked and autoregressive language models (MLMs and ALMs, respectively); phoneme articulation differences in speech models; and distinctions between edges, foreground, and background pixels in image classifiers. After introducing our approach to Shapley interactions, we apply them in a variety of settings and find the following.

- When we control for positional distance (Section 3.1), Transformer-based MLMs—but not ALMs—show a strong correlation between feature interaction and syntactic proximity on a pair of context tokens (Section 3.2). Both LMs often exhibit stronger interactions between pairs of context tokens within an idiomatic Multiword Expression (MWE), but the pattern (Section 3.3) is more consistent in MLMs predicting nearby tokens and in ALMs predicting distant tokens. This combination of observations would suggest that the structure of nonlinear interactions learned by MLMs is more overtly hierarchical than that of ALMs.

- It is known to phonologists that the acoustics of a vowel cannot be interpreted in isolation because the vocal tract is shaped by nearby consonants (Rakerd, 1984). We find that acoustic features in speech models accordingly interact more around transitions between consonants and vowels compared to transitions between two consonants (Section 4.1). Furthermore, consonance have more nonlinear interactions on average between consecutive acoustic features near their transition if the consonant is articulated with a more open oral cavity similar to a vowel, rather than a closed oral cavity (Section 4.2).

- In image classifiers, pixels close to object boundaries exhibit less local interaction, likely because any perturbations are obscured by nearby edge compression artifacts (Appendix A.1).

Considering the edge pixels themselves in detail, edges interact more with nearby foreground object pixels than with other nearby pixels, but interact similarly with all distant pixels (Appendix A.2). We may infer that the boundary of the object is determined by both foreground and edge pixels, but at a distance other interactions take precedence, such as the edge-edge interactions that determine the shape of an edge overall.

## 2 BACKGROUND: SHAPLEY INTERACTIONS

Shapley values are used to attribute decisions to specific features in predictive models, ideally by exhaustively evaluating each possible *coalition* of interacting features. Specifically, the Shapley value of a feature is obtained by computing the difference in a model's output when a feature is included versus when it is withheld from a given set.

Formally, Shapley values are calculated by taking the marginal contribution of a set of target features $A$ to each subset $S \subseteq N \backslash A$, where $N$ denotes the set of all features. For a value function $v$, here the logit outputs of a neural network, the Shapley value is the weighted average across all possible subsets when ablating $A$, given by:

$$\phi(A) = \sum_{S \subseteq N \backslash A} \frac{|S|!(|N| - |S| - 1)!}{|N|!}(v(S \cup A) - v(S)) \tag{1}$$

Shapley values decompose into a close approximation of the output when interactions are additive, such that $\phi(\emptyset) \approx \sum_{i \in S} v(\{i\})$. In scenarios where features are dependent and their composition is non-linear, Shapley values do not account for interacting effects between coalitions. The violation of Shapley assumptions can be computed through the Shapley residual (Kumar et al., 2021):

$$r_i = \nabla_i \phi - \nabla \phi(\{i\}) \tag{2}$$

To calculate second order interactions, we rely on the **Shapley Taylor interaction index** (STII) (Agarwal et al., 2019) using the discrete second-order derivative. For simplicity, we consider the case of interaction between a pair of feature sets $A$ and $B$. Shapley values are scalars, but the influence on each unit in a vector can be computed individually to form a vector of Shapley values. Taking this approach to compute a vector of interactions, we use the norm as a scalar metric of interaction. Similar to Saphra & Lopez (2020), we scale the residual by the norm of the entire sequence with no feature ablations.

$$\text{STII}_{A,B} = \frac{\|\phi(\emptyset) - \phi(A) - \phi(B) + \phi(A, B)\|_2}{\|\phi(\emptyset)\|_2} \tag{3}$$

Calculating the Shapley values for each coalition requires iterating over the powerset of $N$, requiring $O(2^{|N|})$ calculations. In high-dimensional input spaces, the exact calculation of Shapley residuals is therefore prohibitively expensive. We approximate Shapley values by using Monte Carlo Permutation Sampling (Castro et al., 2009).

## 3 LANGUAGE MODELING

Our first experiments are on language models, measuring how known associations between tokens correlate with Shapley-based measures of feature interaction. We consider the influence of token position, idiomatic phrases, and syntax. We find that MLMs and ALMs differ in their interaction structure, especially in how they respond to syntax.

**Datasets** All English language modeling experiments use wikitext-2-raw-v1 (Merity et al., 2016) tokenized and dependency parsed (for syntax experiments) with spaCy (Honnibal et al., 2020). For MWE experiments, we use the AMALGrAM (Schneider et al., 2014a) supersense tagger, which identifies both strong and weak (Schneider et al., 2014b) MWEs. For syntax experiments, we resolve incompatibilities between the spaCy tokenizer and the model-specific tokenizers by assigning overlapping tokens a syntactic distance of zero.

**Experimental Setup** We analyze two models, the ALM GPT-2 (Radford et al., 2019) and the MLM BERT-base-uncased (Devlin et al., 2018). We apply softmax to logit outputs to ensure interactions across examples are comparable. Each sample sentence is unpadded and truncated to 20 tokens. The first and second tokens in an interacting pair are, respectively, $x_{t_1}$ and $x_{t_2}$ where $t_1$ and $t_2$ provide the indices of the features in an input sequence. We measure their non-linear interactions by the Shapley interaction of two feature sets containing a single token each, $\text{STII}_{\{x_{t_1}\},\{x_{t_2}\}}$. We denote the index of the target token to be predicted as $t_{target}$. We ablate the tokens by replacing them with the padding token of the tokenizer.

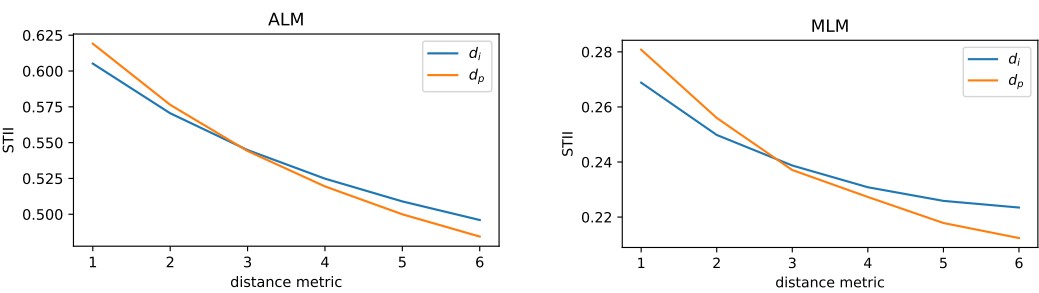

Figure 1: Average interaction scores monotonically decrease with greater interacting token pair distance $d_i$ and with a greater prediction distance $d_p$.

## 3.1 POSITIONAL DISTANCES

One potential factor influencing interactions between tokens is the positional distance between the interacting pair, where it is likely that stronger interactions occur between closer tokens. We also consider the positional distance between the interacting pair and the target of prediction. These surface-level positional differences are the dominant factor in feature interaction.

When relating STII to the latent structure of a sequence, we control for both **interacting pair distance** $d_i$, i.e., positional distance between the interacting pair (Equation 4); and **prediction distance** $d_p$, i.e., the positional distance between the pair and the predicted target token (Equation 5).

$$d_i(x_{t_1}, x_{t_2}, x_{t_{\text{target}}}) = t_2 - t_1 \tag{4}$$

$$d_p(x_{t_1}, x_{t_2}, x_{t_{\text{target}}}) = \min_{t \in \{t_1, t_2\}} |t_{\text{target}} - t| \tag{5}$$

### 3.1.1 RESULTS

We confirm from Figure 1 that in both ALMs and MLMs, STII monotonically decreases at greater distances, whether between the interacting pair (interacting pair distance $d_i$) or between the last token in that pair and the target prediction token (prediction distance $d_p$). The dramatic decline of STII with increased prediction distance implies that when these models predict tokens, they treat the more distant context as a bag of words rather than as complex syntactic relations (Khandelwal et al., 2018). We also see that closer tokens interact more strongly with each other. Our other experiments stratify samples by these distances when we study other factors influencing feature interactions, as it is clear that higher interacting pair distance and positional distance both reduce the interaction score in MLMs and ALMs.

## 3.2 SYNTACTIC DISTANCE

Syntactic structure can also influence an LM's predictions. If a model composed distant syntactic relations in a linear way, it would treat the wider context as though it were a bag of words. By instead exhibiting strong interactions between syntactically close tokens, the model would closely entangle the meaning of a modifier with its head. We measure **syntactic distance** by the number of dependency edges traversed to connect a pair of tokens, a metric encoded by projected representations in both MLMs (Hewitt & Manning, 2019) and ALMs (Murty et al., 2022). We verify the role of

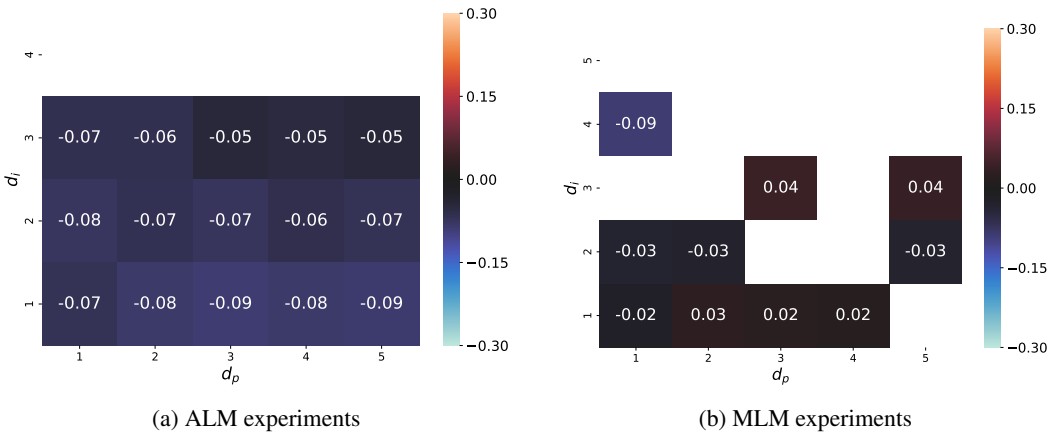

(a) ALM experiments          (b) MLM experiments

Figure 2: Each cell represents the Spearman correlation between syntactic distance and STII, for a given interacting pair distance and prediction distance. Each syntactic distance included must have at least 50 data points. We only provide results for cells where there exists at least one direct syntactic modifier pair separated by the positional distance $d_i$ and the Spearman correlation given at that cell is statistically significance ($p < 0.05$).

modifier connections by the Spearman correlation between syntactic distance and STII, stratified by interacting pair distance and prediction distance.

### 3.2.1 RESULTS

Figure 2 shows correlation between syntactic distance and STII. Our analysis reveals that, for autoregressive language models (ALMs), all statistically significant correlations are negative. In contrast, non-autoregressive language models (MLMs) exhibit both positive and negative correlations. This finding aligns with previous research on LSTMs (Saphra & Lopez, 2020), which indicates that syntax is handled more consistently in autoregressive models, while non-autoregressive models display greater variability.

The inconsistencies observed in non-autoregressive models may stem from their handling of positional proximity in less intuitive ways, complicating the relationship between syntactic and linear distance. The interaction between these two dimensions may be more difficult to manage in MLMs, leading to the varied correlation outcomes.

This finding suggests that we can interpret feature interaction as a distinctly syntactic alternative to the inherent distance encoding found in autoregressive architectures (Haviv et al., 2022). In these models, the degree of interaction is learned to prioritize syntactic relationships rather than depending solely on positional information within the language modeling objective. This highlights a fundamental difference in how these models integrate syntactic structure and distance.

### 3.3 MULTIWORD EXPRESSIONS

Classical treatments of semantics are compositional, implying that the meaning of a sentence is derived by composing the meanings of each individual word. However, there are groups of words whose meaning can only be derived when looking at the entire group rather than the individual words. These word groups, known as **multiword expressions** (MWEs), include idioms like *break a leg*, where the isolated meaning of each of the component words *break*, *a*, and *leg* fail to compose the meaning of the entire expression. Higher interaction values for the tokens in the idiom would indicate a less compositional treatment of the whole phrase.

In these experiments, we compare interactions between arbitrary pairs of tokens to interactions between tokens contained within an MWE. The extreme case where there is no Shapley residual would imply perfect compositionality—after all, linear addition is compositional—so our hypothesis is that MWEs have a larger than average residual.

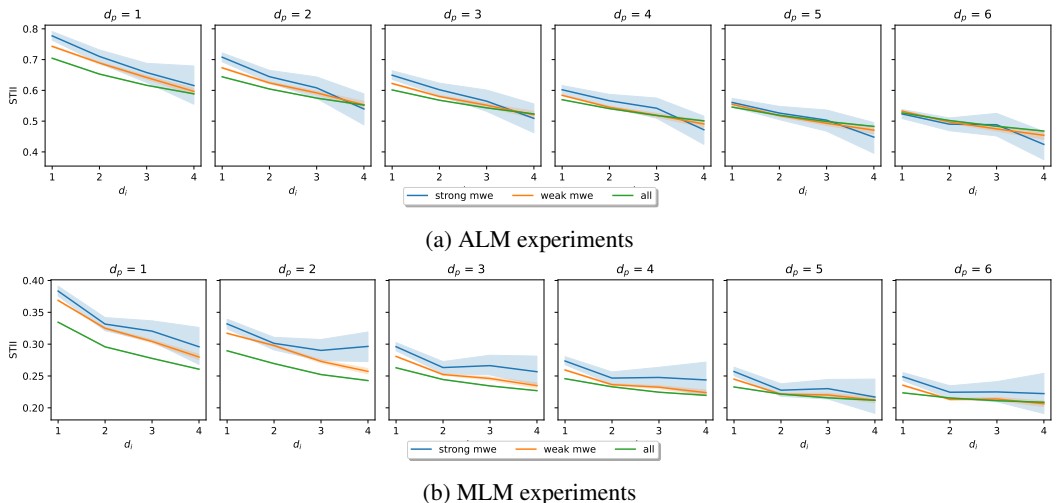

(a) ALM experiments

(b) MLM experiments

Figure 3: Each plot represents the trend in STII for a given prediction distance $d_p$, responding to the x-axis of interacting pair distance $d_i$. Trend is given for the average over: all pairs, strong MWE relations, and weak MWE relations.

### 3.3.1 RESULTS

Figure 3 compares the STII between tokens that belong to the same MWE to the average STII between all tokens, stratified by interacting pair distance $d_i$ and prediction distance $d_p$. For the both ALM and MLM (Figure 3b), STII is higher when the interacting pair is in a MWE. The effect is consistent across positional distances and more pronounced when predicting nearby tokens.

Appendix B adds nuance by contrasting tokens that share a MWE with those sharing a word. Although the MLM increases interaction between MWE tokens, the model decreases interaction between tokens within the same word. This effect is present in multiple languages—even in synthetic languages which compose words from long sequences of morphemes—and our investigation suggests that same-word interaction scores are lower because subword tokens are predictable in context. In contrast, MWEs are often sequences of common words which change meaning within the context of an idiom.

## 4 AUTOMATED SPEECH RECOGNITION

When interpreting interactions in speech models—in this case, predictive models trained on speech—we consider the structure imposed by the phonology that mediates the conversion of intended words to acoustic signal. The study of phonology is based on distinctions between different **phonemes**, such as the sounds represented by *s* or *ee* in English text. These phonemes are the intended sound, but any actual sound produced in context is called a **phone**, and its acoustic features are dictated by the shape of the vocal tract at the beginning and end of its duration, determined by the phones around it.

Because speech is a series of continuous transitions between phones, acoustic features often cannot be interpreted as a specific phoneme without access to their surrounding context. This dependence on context is particularly true for vowels (Rakerd, 1984), as well as for vowel-like consonants such as <l> and <j>. These variations in the dependence between phonemes are what we focus on in our speech experiments.

Rather than considering a single pair of interacting individual input features, we use the average pairwise interaction for all consecutive acoustic features within a time interval. Note that consecutive features can be meaningful ways to study the interaction between phones because the transition between phones is continuous and without a well-defined boundary. For a given interval length, we measure STII between all temporally consecutive features $p_{t_1}$ and $p_{t_2}$ when predicting the immediate next sound $p_{t_3}$. Formally, the interaction $N$ between different phonemes over a temporal interval

within range $\delta$ of the approximated phone boundary time $t_b$ is:

$$\bar{r}_\delta = \sum_{t_1 = t_b - \delta}^{t_b + \delta} \text{STII}_{p_{t_1}, p_{t_2}} \tag{6}$$

Note, however, that in the case where no acoustic feature is sampled at exactly $t_b - \delta$, we instead start the summation with $t_1$ at the earliest timestamp such that $t_1 \geq t_b - \delta$.

As in other experiments, we control for the distance between features as a factor in feature pair interaction. In speech, the relevant distance is the temporal interval between the pair of acoustic features. Features that occur far apart in time should interact relatively little, as the vocal tract will eventually reshape to prepare for the next phoneme. By limiting our analysis only to consecutive acoustic samples, we therefore control for temporal interval distance.

**Datasets** Our speech experiments use the Common Voice dataset (Ardila et al., 2020) of English language voice recordings, which are contributed by volunteers around the world and comprise 92 hours of recorded speech. This compilation is characterized by its rich diversity, featuring a total of 1,570 unique voices. We pre-process the dataset by aligning the audio recordings with their corresponding phonemes using p2fa_py3 [1], an implementation of the Penn Phonetics Lab Forced Aligner (Yuan et al., 2008), which uses acoustic models to map the audio recordings to their corresponding phonemes. We preprocess all audio files to a WAV and standard sampling rate and then use p2fa_py3 to detect and align phonemes within the speech to their corresponding timeframes in the recordings, marking the start and end of each phoneme. It is important to note, as a caveat to the following results, that identifying the exact duration of a phoneme is not only challenging but undefined in practice, as the vocal tract is in a state of continuous transition between phonemes throughout an utterance.

**Experimental Setup** The target of our speech model analysis is the Wav2Vec 2.0 model wav2vec2-base-960h (Baevski et al., 2020), which is trained on 960 hours of English audio to predict the next sound in a recording. When computing Shapley values, ablated acoustic features are replaced with silence.

## 4.1 VOWELS AND CONSONANTS

Vowels are formed with an open vocal tract that produces no turbulent airflow, but the specific position of each part of that anatomy—and consequently the resonance of the cavity—is largely determined by the surrounding consonants. As predicted by the phonology literature (Rakerd, 1984), consonants can therefore often be interpreted in isolation, but vowels rely on nearby acoustic features from surrounding consonants. In Figure 4, we see that the consecutive acoustic features interact more for the 0.15 interval around a consonant-vowel boundary than around a consonant-consonant boundary.

## 4.2 MANNER OF CONSONANT ARTICULATION

While vowels are defined by how open and fronted the vocal tract is during formation, pulmonic consonants are defined by three main features: place of articulation (such as at the teeth), voicing (the difference between  and <z>), and manner of articulation. The manner of articulation defines a hierarchy of tract occlusion from the stops (short consonants formed closing the oral cavity completely) to the approximants (consonants that produce only slightly more turbulent airflow than a vowel). Therefore, some consonants in practice behave more like vowels, and we expect them to exhibit more nonlinear interactions across phoneme boundaries, as vowels do.

Our hypothesis is confirmed in Figure 5, modeled on a traditional consonant International Phonetic Alphabet chart where each phoneme is placed in a cell with its column specified by location and row specified by manner of articulation, with one side of a cell for voiced variants and the other for unvoiced variants. Although the pattern is not perfect—highlighting <w> as an approximant that is unusually interpretable in isolation—the figure shows high cross-phoneme STII for more sonorant consonants on the lower rows, which are articulated like vowels with a more open oral

---

[1] https://github.com/jaekookang/p2fa_py3

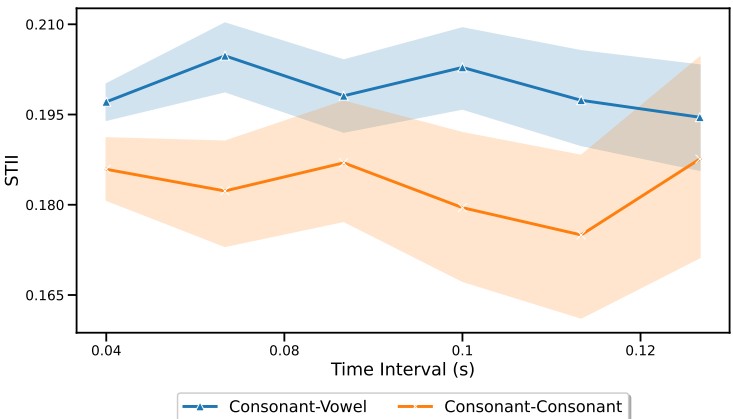

Figure 4: Comparison between average vowel-consonant interaction and consonant-consonant interaction for acoustic features from adjacent phonemes. Temporal distance is considered, but unlike in other tasks, the relationship between pair distance and pair interaction is not generally monotonic. Confidence intervals are provided by bootstrap.

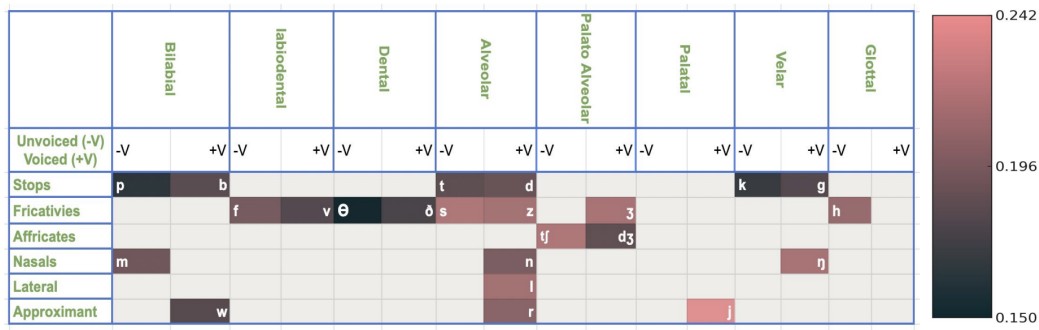

Figure 5: Consonant chart with a heat map indicating average interaction with acoustic features from adjacent phonemes. Is is traditional in phonology, the column describe the place of articulation within the vocal tract while the row describes the manner of articulation. Only interactions for acoustic features within 0.1s range around the phoneme boundary are considered.

cavity. Conversely, many of the phonemes which can be interpreted in isolation fall on the top rows, where the cavity is more closed, producing a phone much quieter than the sound of a vowel.

## 5 RELATED WORK

The basic version of the Shapley value and its approximation techniques do not account for linear and non-linear interaction effects between groupings of features. In a non-linear value function like a modern neural network, therefore, the Shapley value is by definition unfaithful. The Shapley interaction value began with Owen (1972), studying the multilinear extension of Shapley values. Grabisch & Roubens (1999) axiomatizes the Shapley interaction index. Recently, Fumagalli et al. (2023) generalizes an unbiased form for approximation of interactions indices via a sampling-based estimator. Kumar et al. (2021) characterize the limits of Shapley values and provide a geometric interpretation for calculating interactions across all features. Tsai et al. (2023) proposes a generalized approximation of the Shapley value to interactions by extending linear to polynomial approximations, eliminating the need for additional conflicting axioms. Since our work focuses on pairwise interactions in a large feature space, we use the Shapley-Taylor interaction (STII) (Agarwal et al., 2019).

Interpretability research in NLP has long exploited the underlying structure of natural language text.Warstadt et al. (2018) assess the proficiency of language models in recognizing proper grammar

and syntax within sentences through diagnostic challenge sets. Another common approach is probing models to evaluate the incorporation of syntax into language models, proving beneficial but necessitating additional parameters in the modeling process (Hewitt & Manning, 2019; Belinkov, 2021). Bai et al. (2021) consider how to explicitly integrate syntactic information into pre-trained models, providing marginal improvements over the state-of-the-art. Our work adds to a strong body of evidence that LMs to represent the latent structure of language internally.

Our work is also not the first to use feature interactions to investigate model behavior in the context of natural language structure. Prior work in the area focus primarily on older architectures such as LSTMs (Saphra & Lopez, 2020) and simple tasks such as text classification (Jumelet & Zuidema, 2023; Chen et al., 2020; Singh et al., 2019). Building upon the foundational contributions of prior studies, our initial exploration in language models delves into a comprehensive analysis of modern Transformer-based autoregressive and masked language models. In a departure from previous research, we not only examine the relationship between feature interactions and syntax, but also consider their implications for idiomatic expressions.

Moreover, in the scope of analyzing learning algorithms for speech, Chrupała et al. (2020) examined the phonological representation in neural networks trained on spoken languages using probes and representation similarity analysis, revealing phonological information implicitly encoded by internal representations. Furthermore, Markert et al. (2021) analyzed models trained for Automatic Speech Recognition through various attribution methods such as SHAP. These prior works underscore the significance of analyzing low-level feature interactions in models across different modalities, providing motivation for our exploratory study.

## 6   FUTURE WORK

Our primary objective in this work has been to showcase the versatility of these methods while demonstrating how much their interpretation requires a deep understanding of the underlying structure of the data. This work suggests a number of open questions and follow-up problems in addition to its application on further tasks and domains.

Speech has multiple layers of structure, as it comprises both an acoustic signal and the language structure underlying the utterance. Our investigation of feature interactions is limited to the phonetic level, but future work may find the degree to which these multiple layers of linguistic structure affect nonlinear feature interactions. Do these speech models exhibit similar interaction patterns to the autoregressive language models we also analyze? Speech, often neglected in interpretability research, is ripe with open problems.

While we compare the behavior of the models trained on the MLM and ALM objectives, we do not compare any models that are trained on the same objective with different architectures. The inductive bias and function of a given architecture are matters of great interest to many researchers in machine learning, and we believe that measuring nonlinear interactions can provide many insights into how specific models are similar and different.

This work has not taken full advantage of the versatility of Shapley residuals as a tool. Higher order Shapley interactions (Sundararajan et al., 2020) provide a method of hierarchical clustering on features and introduce yet more nuance into approximations of linear and nonlinear behavior in neural networks. We also do not consider interactions of internal model features. We suggest that future work in the area should incorporate knowledge about the underlying semantics of the input as well as the model architecture.

Finally, and most crucially, we believe that followup work in this area should be interdisciplinary. Speech, language, image processing, and other areas that can benefit from interpretability are all well-studied, with decades or even centuries of scientific research. By collaborating with specialists in these data domains, we can potentially contribute not only to the understanding of artificial models, but also to the understanding of the natural phenomena in question. Interpretability is an important new area in the emerging field of AI for scientific understanding and discovery, and we encourage others to start future work by finding domain experts to choose questions worth asking.

## 7 CONCLUSIONS

In accordance with The Bitter Lesson (Sutton, 2019), researchers and engineers typically apply machine learning methods generically, incorporating as little explicit data structure as possible. However, The Bitter Lesson does not apply to *interpretability*. Instead, meaningful interpretations of representational and mechanistic structures at scale should be informed by the underlying structure of data. Our results show how to use constituents, phones, and object boundaries to build a scientific understanding that goes beyond intuitions about n-grams, acoustic features, and pixels.

These results have spanned modality and task. By measuring feature interaction in language models, we find that multiword expressions are handled well compositionally both in MLMs and ALMs. In speech prediction models, we show that consecutive acoustic features near a phone transition have more nonlinear interactions if the transition is between a consonant and vowel, rather than between two consonants. We also see that in this sense, sonorant consonants behave more like vowels. In our image classifier experiments, we see that pixels on object boundaries interact most with nearby pixels in the object foreground, but interact similarly with all pixels further away. Furthermore, pixels closer to an object boundary are more locally linear, indicating that they have little individual influence on the semantics of their local region.

These studies do not focus on individual data samples, but on patterns in the structure underlying the data. Understanding these general patterns requires greater domain expertise than is often required for sample-level interpretability research. We hope to inspire future interdisciplinary work with phonology, syntax, visual perception, and other sciences that characterize corpus-wide structural phenomena.

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

# A   IMAGE CLASSIFICATION

While LMs use hierarchical syntactic structure and speech models interpret continuous transitions between phonemes, image classification models are often viewed as composing high- and low-frequency features (Schubert et al., 2021). Work in computer vision and interpretability often contrasts the lowest frequency features (shapes) with the highest frequency (textures) by localizing them to edge and non-edge pixels, respectively. We will leverage this framework by focusing on the interactions between pixels in the foreground, background, and object boundaries.

We find that edge pixels—associated with the lowest frequency features—have the smallest nonlinear interaction with nearby pixels. Additionally, we compute the magnitude of interactions between pixels at various Manhattan distances that occur in edges, backgrounds, and object foregrounds. We find that edge pixels interact most strongly with nearby pixels if they belong to the object foreground, but this trend does not apply between more distant pixels.

**Datasets**   We use the MNIST handwritten image dataset (LeCun et al., 2010) and the CIFAR-100 (Krizhevsky, 2009) image classification dataset.

**Experimental Setup**   We conduct experiments using pre-trained Vision Transformer (ViT) (Dosovitskiy et al., 2020) models for both MNIST and CIFAR-100 datasets from the Hugging Face model hub. We take a sample of 100 images from different classes in each dataset for our interpretability analysis target. The outputs are standardized at an image level to control for the variance in the interaction level across different images. We use a reference value of the channel-wise mean pixel value to replace ablated features. We evaluate other options for a reference value and include these in the Appendix.

## A.1   LOCAL RECONSTRUCTION

For each pixel, we consider a local square grid of length $d$ and take the mean value of the neighboring pixel interactions according to Equation 7, excluding the central pixel at $(i, j)$ from the mean.

$$\bar{r}_{ij} = \frac{1}{N} \sum_{x=i-d}^{i+d} \sum_{y=j-d}^{j+d} \text{STII}_{(i,j),(x,y)} \tag{7}$$

By taking an aggregate of the interactions in a small neighborhood of pixels, we reproduce a version of the image (Figure 6) using a heatmap. The similarity to the original image is predominantly because edges have much lower average interactions with neighboring pixels than does the object or background. Within the object foreground, furthermore, the farther the pixel is from the nearest edge, the higher the interaction values for that pixel. Pixels near the edge have smaller interaction values because the local region's structure is defined by the edge, so perturbing a nearby pixel does not change the patch's overall interpretation and the perturbed pixel blends in as an edge compression artifact. In contrast, perturbing a pixel far from the edge changes the potential semantics of the local region.

## A.2   PAIRWISE PIXEL INTERACTIONS

Each pixel in an image falls within an edge, object foreground, or background region. We now consider the interactions between pixels on the edge of an object and those elsewhere in an image: elsewhere on an edge, on the object foreground, or in the background. As the strength of interaction is strongly correlated with the distance between target pixel pairs, we control for their Manhattan distance.

For both the CIFAR-100 and MNIST datasets, close pixels have the highest interaction with an edge pixel if they are in the object foreground distance, possibly due to the model inferring the boundary of the edge. As the distance between interacting pixels increases, all interaction values fall; farther pixels appear to contribute similarly to the inferred edge boundary regardless of where they fall in the underlying scene.

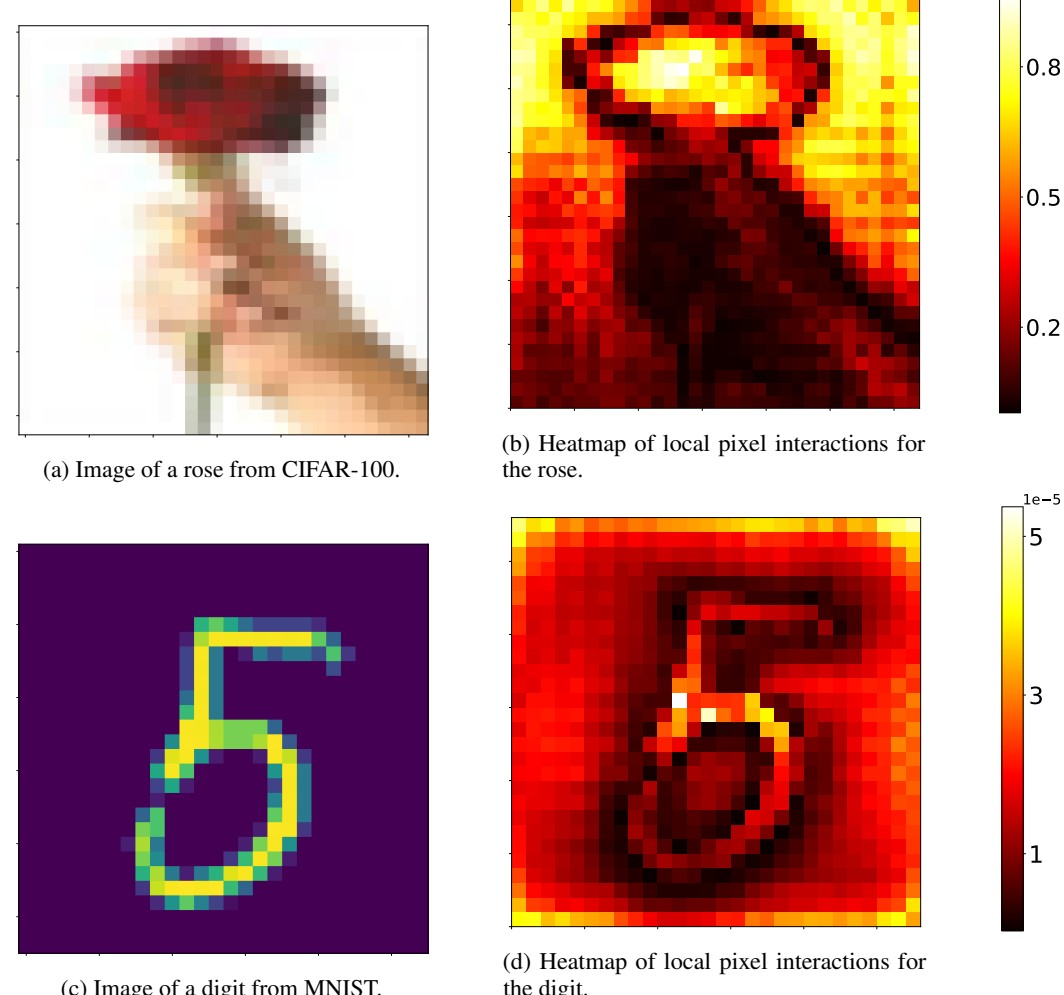

(a) Image of a rose from CIFAR-100.

(b) Heatmap of local pixel interactions for the rose.

(c) Image of a digit from MNIST.

(d) Heatmap of local pixel interactions for the digit.

Figure 6: Each pixel represents the average interaction with nearby (distance = 2) pixels as calculated by Equation 7.

## B  MULTILINGUAL SUBWORD TOKENS

We present additional experiments conducted in a multilingual setting, aiming to explore the distinctive nature of relationships between subword tokens that belong to the same word. We ran the experiments on German, Turkish and English. We observe that on average STII is lower for subword token pairs from the same word, as compared to STII for all adjacent pairs. The possible reason for that is that a subword token is more predictable from the rest of the word, so it is easier for the model to identify the possible token if it belongs to the same word. The interactions are therefore dampened because the addition of the masked subword token contains very little information that cannot be inferred from context.

**Datasets**  We use the Turkish NER dataset ((Altinok, 2023)) and the GermEval German NER dataset ((Benikova et al., 2014)). As in English, these datasets are drawn from Wikipedia.

**Experimental Setup**  We use XLM-RoBERTa, a multilingual variant of RoBERTa designed for MLM modeling. Non-linear interactions are then calculated for same-word subword tokens and interactions between any two tokens. The methodology mirrored that described earlier for calculating non-linear interactions, specifically the MWE experiments.

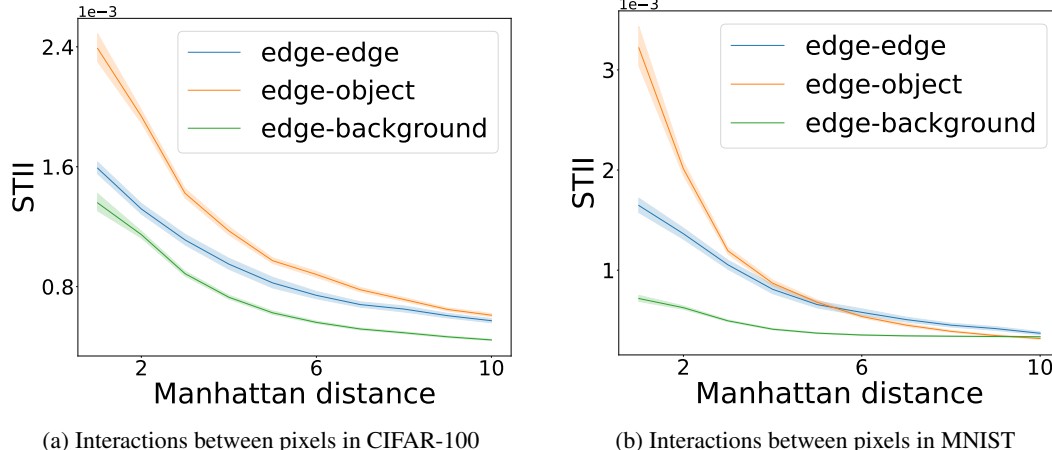

(a) Interactions between pixels in CIFAR-100

(b) Interactions between pixels in MNIST

Figure 7: Each line represents the average interactions between features in the image as measured by STII, controlling for Manhattan distance. The error bar corresponds to a 95% confidence interval obtained via bootstrap sampling.

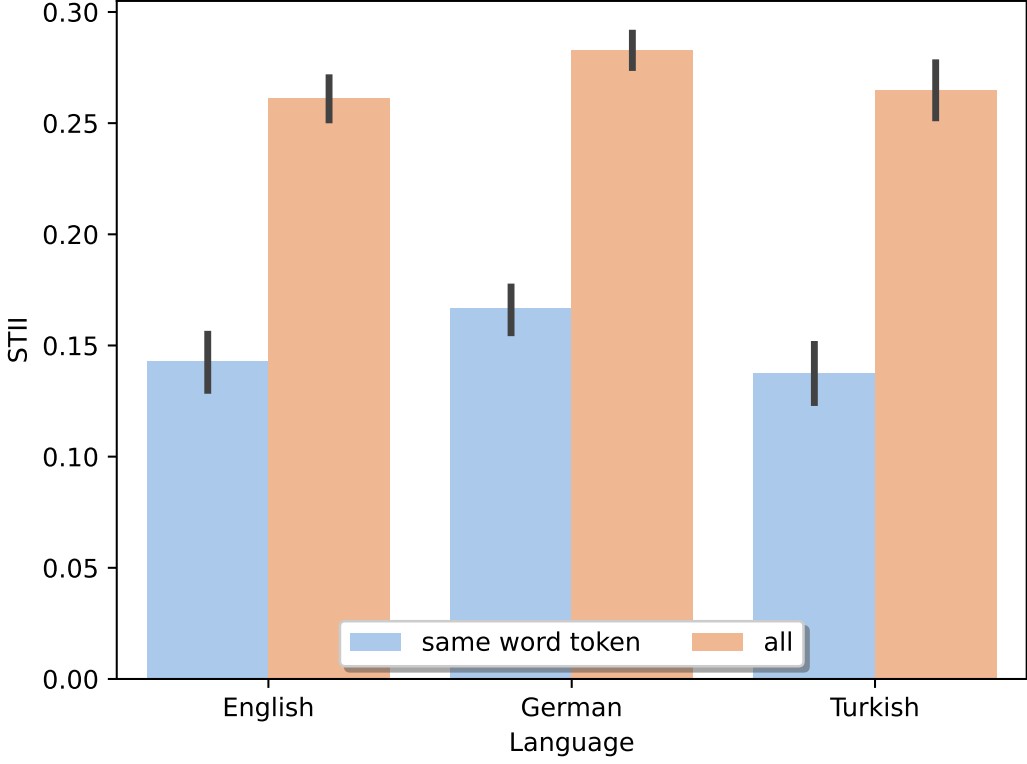

Figure 8: Each plot represents the trend in STII for adjacent context tokens when predicting the immediate next token. Trend is given for the average over all pairs and tokens within the same word

## B.1 RESULTS

Figure 8 illustrates significantly lower Shapley interaction values for tokens within the same word compared to interactions between all token pairs. Perhaps surprisingly, this effect holds for both diverse morphological types, meaning that these languages are extremely different in how they construct words. Turkish, as an agglutinative language, would have many subword tokens with varied inflections; German, a polysynthetic language, has many subword tokens that correspond to their own

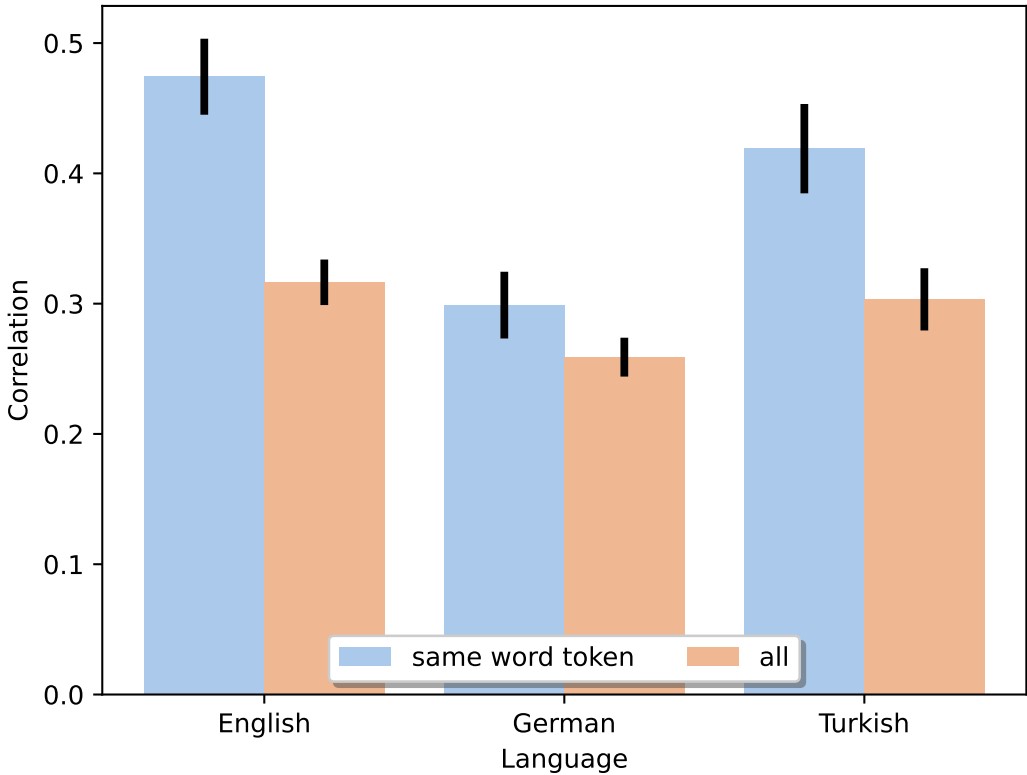

Figure 9: Correlation of loss with STII

words in other languages; and English, a moderately analytic language, would have fewer distinct morphemes per word. However, tokenization schemes do not necessarily align with morphological distinctions, and so they may be inferred from context even in a synthetic language such as Turkish or German.

Now, we consider the hypothesis that the greater predictability of subword tokens is the reason why same-word tokens have lower interactions. Examining the Spearman correlations of cross-entropy loss with Shapley values for tokens within the same word versus any pair of tokens (Figure 9), we observe a positive correlation in both cases. This suggests that as the model's uncertainty in predicting a token increases, non-linear interactions tend to elevate. Intriguingly, the correlation for tokens within the same word surpasses the average case, indicating a more pronounced increase in interaction—and same-word interaction as a case to be further considered.

The results differ from the MWE case, where the interactions are much stronger within the MWE, here the interactions are lesser within the tokens of the same word, likely because subword tokens are more predictable given the rest of the word than the tokens in MWE, which are sequences of common words that often change meaning in the context of a specific idiom.

## C    CHOICE OF REFERENCE FOR IMAGE CLASSIFICATION

The reference value refers to the choice of substitution value for a given pixel in an image. A reference value is needed for the evaluation of the Shapley value, as the value substituted for an ablated feature.

We evaluate several options for the reference value to use for pixels $A, B$ in Equation 3 when ablating features from an image. In practice, choosing a reference value should incorporate domain specific knowledge (Shrikumar et al., 2017). In our experiments we consider zero-reference and mean-reference, where the ablated pixel values are set to zero or the channel-wise mean value. We

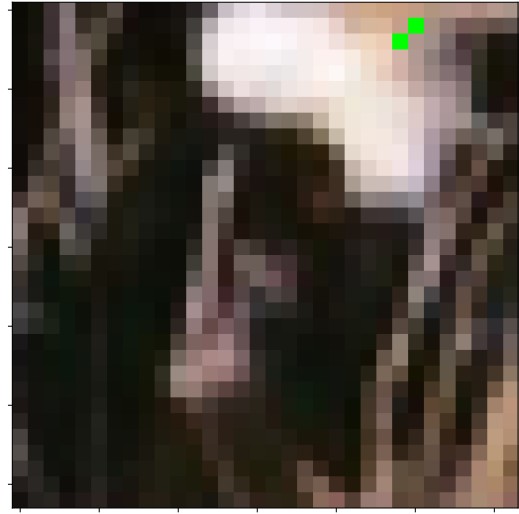

Figure 10: The highlighted pixels on the CIFAR image both change the prediction from true class, skunk, to a mouse when either pixel is changed to the reference value. When both pixels are changed to the reference value, the prediction again becomes skunk. This is likely due to the single black pixel representing an eye feature on the skunk fur.

also evaluate a blur-reference, but due to it's dependence on a local kernel, we decide against this approach because we aim for standardization in the reference value for the pixel pairs.

We recognize that the choice of reference value can impact different datasets in different ways depending on the context. For example, due to MNIST images having a black background, the zero reference will have zero impact when the feature ablation is contained in the background. This contrasts with CIFAR-100 images, where a black pixel can represent an eye feature, thus changing the dynamics of the interaction. An example of an image where ablation forms a possible spurious eye is illustrated in Figure 10.

Although zero-reference and mean-reference approaches yield similar results in our experiments, we opt for the mean reference. This choice is driven by its consistent behavior across datasets, which reduces the chances of having an outlier-like impact on the calculation.

