# OpenReview forum: "Structural Probing with Feature Interaction"
_ICLR.cc/2025/Conference — ICLR 2025 Conference Withdrawn Submission_

### Official Review · Reviewer_EPqW · 2024-11-03

**Soundness:** 2
**Presentation:** 3
**Contribution:** 1
**Rating:** 3
**Confidence:** 4

**Summary:**

Interpretability of representations and models is a critical research problem. This paper uses Shapley Taylor interaction indices (STII) to analyze model representations across language modeling, speech recognition, and image classification. The authors observed some intuitive findings based on STII on these tasks and modalities.

**Strengths:**

(1)	Studying interpretability of representations and model architectures is a valuable research topic.

(2)	It is interesting to see the effectiveness of STII across positional distance, syntactic distance, and MWE in language modeling, speech modeling, and image processing.

**Weaknesses:**

(1)	The most crucial concern of the paper is that this is still an initial stage of an interesting research direction. It is interesting to see the findings of using STII to analyze the impact of representations and model architectures on these text/speech/image tasks. The findings are intuitive, but also overlap with previous findings, for example, prior works on probing representational capabilities of MLMs, CLMs,  and speech representation learning. The paper also lacks comparison of using STII for interpretability analysis with other approaches for interpretability. Moreover, the work lacks diving deeper into the interpretability results and proposing and studying more effective approaches to improve performance on unimodal and multimodal tasks.

(2)	The paper needs to be self-contained w.r.t. important technical details.

a.	Section 2, it is important to at least briefly describe the Monte Carlo Permutation Sampling approach to approximate Shapley values in high dimensional input spaces, instead of just providing a citation.

(3)	The designs of empirical validations are insufficient in some places.

a.	Regarding language modeling, to study the influence of token positions, it is important to cover short to long distances, such as up to the max sequence length 512 tokens. However, in experimental setup, each sample sentence is unpadded and truncated to 20 tokens, and Figure 1 only shows distance metric up to 6. This is quite limited.

(4) As mentioned in Weakness 1, the paper lacks comparisons and analyses with other interpretability works. For speech modeling, as this paper focuses on study of phonology. It would be quite useful to compare STII to other speech representations that are also found to correspond to discrimination of phonemes.

**Questions:**

Please check the points listed under Weaknesses.

---

### Official Review · Reviewer_HQM4 · 2024-11-04

**Soundness:** 2
**Presentation:** 1
**Contribution:** 2
**Rating:** 3
**Confidence:** 3

**Summary:**

The paper utilizes Shapley Taylor Interaction Indices (STII) to analyze feature interactions in neural networks involving language, speech, and image modalities.

**Strengths:**

1. Unlike previous studies in the area, this paper performs a feature attribution analysis of modern Transformer-based autoregressive and masked language models.

**Weaknesses:**

1. The problem statement and motivation of the work are not clear enough. In the introduction, the paper only mentions that feature interaction helps with interpreting neural networks, with no relevant references. This is insufficient to motivate and justify the problem statement. It would be helpful to mention whether the mentioned problem is widely studied, and what the application-specific benefits of studying feature interaction are, with appropriate examples and citations.

2. The proposed analysis method (Shapley Taylor Interaction Index) is mentioned without any relevant context, and it is unclear how this method compares with prior methods, if any. Is STII considered a better method for studying feature attribution in neural networks? If so, could you explain how it is better? Do you have any intuition on why you consider STII a reasonable candidate over other methods? Was your decision influenced somehow by a new challenge posed by a target architecture such as Transformer?

**Questions:**

See the weakness for relevant questions.

---

### Official Review · Reviewer_nLwq · 2024-11-05

**Soundness:** 3
**Presentation:** 3
**Contribution:** 3
**Rating:** 6
**Confidence:** 1

**Summary:**

The paper investigates feature interactions in neural networks through the lens of Shapley Taylor Interaction Indices (STII), with a particular focus on understanding the non-linear feature interactions across language, speech, and image classification tasks. The study spans various model types, including language models (MLMs and ALMs), speech models, and image classifiers, to highlight how underlying data structures affect feature interaction patterns.

**Strengths:**

- The paper’s application of Shapley Taylor Interaction Indices (STII) to idiomatic expressions in language models is a novel contribution, revealing how models capture non-compositional patterns in multiword expressions and providing unique insights into how these expressions differ from regular syntax
- Covering multiple domains (language, speech, and image) is a unique approach that expands the paper's impact across NLP, speech, and computer vision communities, particularly in the interpretability of domain-specific structures.

**Weaknesses:**

- The paper is dense with technical details, making it challenging to follow at times, particularly with abbreviations. Some abbreviations, such as MLM, ALM, and STII, are introduced early but only explained in detail later in the paper. A refining pass to ensure that terms and abbreviations are defined when first mentioned would improve readability and accessibility for a broader audience.
- Calculating Shapley values and interactions in high-dimensional spaces is computationally intensive. While the paper briefly addresses this by using an approximation method, further insights into the practicality of applying STII on large-scale datasets would be valuable. Such details could help clarify the method's feasibility for broader applications where computational resources are a key consideration.

**Questions:**

- Could the authors please comment on the observed scales of STII values across different experimental settings? For instance, STII values for MLM experiments range from 0.4 to 0.8, while those for ALM experiments are between 0.2 and 0.4. In contrast, the STII values in the phoneme experiments have a smaller range, from 0.165 to 0.210. Could the authors provide insights into what might account for these variations in scale? Specifically, in the case of the phoneme experiments, might the narrower range be influenced by the Monte Carlo sampling process used to approximate Shapley values?

---

### Official Review · Reviewer_XCx5 · 2024-11-05

**Soundness:** 2
**Presentation:** 2
**Contribution:** 1
**Rating:** 1
**Confidence:** 4

**Summary:**

The authors propose to use Shapley-Taylor Interaction Index to investigate whether the value function as represented by masked or autoregressive LMs mirrors known patterns in the data, specifically close associations between text tokens in multi-word expressions or positionally close/distant tokens, vowel-consonant vs. consonant-consonant transitions, and   pixel interaction near object boundaries in image classifiers. The authors find that the expected patterns are indeed reflected in  the STII values and that autoregressive LMs differ from masked prediction LMs, suggesting that the latter learn more of a hierarchical structure.

**Strengths:**

The authors study data patterns in three different modalities.

**Weaknesses:**

1. It is not quite clear to me what the study contributes - the question seems to be 'can STII give some indication of the feature interactions in the data', but this is not further motivated. Why are Shapley interactions the method of choice? How were the data patterns (MWE etc.) chosen - it seems that these were handpicked. What difference in STII is significant, and is it a reliable tool to evaluate different models?
2. The method is extremely computationally complex. If data analysis is the goal, (conditional) mutual information or canonical correlation analysis can be used instead -- these  have been used before for similar problems. Authors should take this into account and provide a well-reasoned analysis why a new technique is needed.
3. A large part of the analysis (the entire 3rd application) is in the appendix. It might have been better to condense the paper and provide just the main results.

**Questions:**

I would clearly frame the research hypothesis to be investigated and provide baseline numbers from tried-and-true techniques like mutual information or CCA. The application confirming known structures in the data does not add any new insights - I would either try to use this technique to discover new dependencies in the data that may not have been obvious, or compare a broader range of models. If the latter, SOTA models should be used, and differences should be compared quantitatively, with estimates of statistical significance.

---

### Note · Authors · 2024-12-04

**Comment:**

Thank you to all reviewers for your suggestions on improving the paper.

**Withdrawal Confirmation:**

I have read and agree with the venue's withdrawal policy on behalf of myself and my co-authors.